# Implementation of Artificial Intelligence in Personalized Prognostic Assessment of Lung Cancer: A Narrative Review

**DOI:** 10.3390/cancers16101832

**Published:** 2024-05-10

**Authors:** Filippo Lococo, Galal Ghaly, Marco Chiappetta, Sara Flamini, Jessica Evangelista, Emilio Bria, Alessio Stefani, Emanuele Vita, Antonella Martino, Luca Boldrini, Carolina Sassorossi, Annalisa Campanella, Stefano Margaritora, Abdelrahman Mohammed

**Affiliations:** 1Faculty of Medicine and Surgery, Catholic University of Sacred Heart, 00168 Rome, Italy; marco.chiappetta@policlinicogemelli.it (M.C.); jessica.evangelista@policlinicogemelli.it (J.E.); emilio.bria@policlinicogemelli.it (E.B.); alessio.stefani@unicatt.it (A.S.); emanuele.vita@guest.policlinicogemelli.it (E.V.); antonella.martino@policlinicogemelli.it (A.M.); luca.boldrini@policlinicogemelli.it (L.B.); carolina.sassorossi01@unicatt.it (C.S.); stefano.margaritora@policlinicogemelli.it (S.M.); 2Thoracic Surgery, A. Gemelli University Hospital Foundation IRCCS, 00168 Rome, Italy; sara.flamini@guest.policlinicogemelli.it (S.F.); annalisa.campanella@guest.policlinicogemelli.it (A.C.); 3Faculty of Medicine and Surgery, Thoracic Surgery Unit, Cairo University, Giza 12613, Egypt; galal.ghaly@nci.cu.edu.eg (G.G.); filippomaria.lococo@unicatt.it (A.M.); 4Medical Oncology, A. Gemelli University Hospital Foundation IRCCS, 00168 Rome, Italy; 5Radiotherapy Unit, A. Gemelli University Hospital Foundation IRCCS, 00168 Rome, Italy

**Keywords:** NSCLC, artificial intelligence, machine learning, lung cancer surgery, prediction model radiomics

## Abstract

**Simple Summary:**

Artificial intelligence (AI) has largely changed the overall management of non-small-cell lung cancer (NSCLC) by enhancing different aspects, including staging, prognosis assessment, treatment prediction, response evaluation, recurrence/prognosis prediction, and personalized prognostic assessment. In the present narrative review, we analyzed and discuss studies reporting on how AI algorithms could predict responses to various treatment modalities, such as surgery, radiotherapy, chemotherapy, targeted therapy, and immunotherapy.

**Abstract:**

Artificial Intelligence (AI) has revolutionized the management of non-small-cell lung cancer (NSCLC) by enhancing different aspects, including staging, prognosis assessment, treatment prediction, response evaluation, recurrence/prognosis prediction, and personalized prognostic assessment. AI algorithms may accurately classify NSCLC stages using machine learning techniques and deep imaging data analysis. This could potentially improve precision and efficiency in staging, facilitating personalized treatment decisions. Furthermore, there are data suggesting the potential application of AI-based models in predicting prognosis in terms of survival rates and disease progression by integrating clinical, imaging and molecular data. In the present narrative review, we will analyze the preliminary studies reporting on how AI algorithms could predict responses to various treatment modalities, such as surgery, radiotherapy, chemotherapy, targeted therapy, and immunotherapy. There is robust evidence suggesting that AI also plays a crucial role in predicting the likelihood of tumor recurrence after surgery and the pattern of failure, which has significant implications for tailoring adjuvant treatments. The successful implementation of AI in personalized prognostic assessment requires the integration of different data sources, including clinical, molecular, and imaging data. Machine learning (ML) and deep learning (DL) techniques enable AI models to analyze these data and generate personalized prognostic predictions, allowing for a precise and individualized approach to patient care. However, challenges relating to data quality, interpretability, and the ability of AI models to generalize need to be addressed. Collaboration among clinicians, data scientists, and regulators is critical for the responsible implementation of AI and for maximizing its benefits in providing a more personalized prognostic assessment. Continued research, validation, and collaboration are essential to fully exploit the potential of AI in NSCLC management and improve patient outcomes. Herein, we have summarized the state of the art of applications of AI in lung cancer for predicting staging, prognosis, and pattern of recurrence after treatment in order to provide to the readers a large comprehensive overview of this challenging issue.

## 1. Introduction

Lung cancer remains a significant and challenging disease, being the leading cause of cancer-related death worldwide. Late diagnosis and the inherent variability in the imaging characteristics and histopathology of lung cancer present significant barriers for clinicians in determining the optimal treatment approach, which are now the main limits to the optimization of the treatment. Non-small-cell lung cancer (NSCLC) is the predominant subtype and its treatment typically involves surgery, radiotherapy, chemotherapy, immunotherapy or molecular targeted therapy [1]. Despite remarkable advances in lung cancer treatment, overall survival remains disappointingly low, necessitating the use of personalized strategies to improve patient outcomes. Advances in precision medicine and biomarker identification offer hope to improve patient outcomes. Clinicians must adopt a personalized approach to treat lung cancer patients to improve the efficacy of current treatment options. In this framework, to answer to several unmet needs of clinicians, radiomics approaches may play a crucial role. Radiomics is an emerging and rapidly developing field that integrates knowledge from radiology, oncology, and computer science, emphasizing the integration of medicine and engineering. In recent years, AI has emerged as a promising tool in the field of oncology, particularly in image recognition and analysis. This new branch may help in the precise recognition of lung nodules thanks to well-defined radiological features and may lead to a more accurate diagnosis. This would reduce the limited capability of the human eye to predict the kind of lesion, starting only from the images, and would increase the chances of a correct preoperative diagnosis and, as a direct consequence of this, a more customized treatment. Furthermore, where the molecular mutations that drove the tumor growth could be also predicted, targeted therapy could be administrated without any delay. 

There is a shared impression in the scientific community that the future of lung cancer treatment lies in developing innovative approaches that consider the heterogeneity of the disease and, therefore, allow the personalized treatment for each lung cancer patient. 

Artificial Intelligence (AI) is the term used to describe the use of computers and technology to simulate intelligent behavior and critical thinking comparable to that of a human being. John McCarthy first described AI in 1956 as the science and engineering of making intelligent machines. AI is a branch of computer science that consists of a set of algorithms that can interpret large amounts of data to perform complicated tasks and mimic human intelligence [2]. Machine learning (ML) is a type of AI that develops algorithms based mainly on pre-defined existing data without explicit programming. Deep learning (DL) is a sub-discipline of ML based on a neural network structure inspired by the human brain. DL algorithms do not need to pre-define features and can, therefore, learn features directly by navigating the data themselves. This data-driven mode makes them more informative and practical. Convolutional neural networks (CNNs), recurrent neural networks (RNNs), Restricted Boltzmann Machines (RBMs) and Autoencoders (AE) are the most used models of machine learning in the field of medical images. They are also the most popular types of DL architecture in medical image analysis [3]. There is growing evidence that radiomics can be used to quantitatively characterize tumors for tasks such as disease characterization or outcome prediction [4]. AI is aiming to transform medical practice, particularly in oncology, and has recently surpassed previous related technologies in image recognition and analysis. AI can extract all kinds of information from the image that is generally ignored by the human operator [5]. There is growing evidence suggesting that radiomics can be used to quantitatively characterize tumors for tasks such as disease characterization or outcome prediction. This represents an important research direction in medical applications [5,6].

## 2. Methodology of Research

This narrative review is a summary of relevant publications found through a literature search performed in the PubMed Library database (see Figure 1). 

Comprehensive research of the literature was conducted on PubMed from 2007 to 2023. The advanced tool for titles and abstracts was used with the keywords: “AI and Lung nodules”, “Lung cancer and Artificial Intelligence”, “NSCLC and AI”, and “predictive model and NSCLC”. 

Year of publication: Any publication date starting from 1 January 2007 was eligible until 31 October 2023. Language: Only studies with their full text in the English language were included.

Type of study: Only peer-reviewed publications reporting data from at least 20 cases were considered. Secondary expert opinions were excluded at the screening stage. Only full manuscripts were eligible, excluding conference abstracts and proceedings. No constraints were imposed based on the level of evidence.

We included (in the first selection) all original studies describing the use of Artificial Intelligence, radiomics, deep learning, machine learning, lung cancer, lung malignancy, lung nodules, and NSCLC.

A literature analysis was performed by two independent authors (F.L. and G.G) with great experience (>10 years) in both clinical and research activities.

Discrepancies were resolved by a third author (M.C., with about 8 years of experience in surgical/research activities in the field of lung cancer). 

A fourth reviewer, C.S., with specific knowledge of the PRISMA methodology, supervised the overall literature revision process according to the selection criteria reported above. Finally, the articles were thoroughly examined, processed, and summarized according to the goals of the review. Finally, all authors read the selected papers and engaged in discussions to formulate reliable conclusions.

As reported in the flow chart (Figure 1), a total of 295 papers were identified with this search strategy. Out of them, 164 were excluded because of the wrong study design, 69 because of the wrong publication type, and 4 because they were not written in English. In the end, 58 articles were selected, because they were consistent with the aim of the review. 

## 3. Artificial Intelligence and NSCLC Staging (TNM Stage)

The universally accepted staging system for lung cancer relies on the TNM system or the assessment and definition of components associated with the primary neoplasm (T), lymph node involvement (N), and metastatic presence (M). Precise staging of lung cancer holds paramount importance as it forms the cornerstone for therapeutic decisions and prognostic evaluations. Presently, the TNM system for lung cancer has been revised to its eighth edition. Non-invasive techniques (CT, MRI, PET, trans-tracheobronchial and transesophageal ultrasound) must be combined with invasive methods (mediastinoscopy, mediastinotomy needle aspiration, transbronchial biopsy, video thoracoscopy, thoracotomy) for pinpointing the location of diverse lymph node stations. Notably, in contemplating potential multimodal treatment approaches, relying solely on lymph node size criteria for staging is unjustifiable. These criteria, stemming from both CT (computed tomography) and MRI (magnetic resonance imaging), remain unreliable: indeed, it is common to encounter small lymph nodes harboring metastases or large lymph nodes devoid of neoplastic infiltration. The significance of mediastinal examination in clinical staging has spurred research endeavors aimed at identifying the most dependable instrumental method. Recent publications [7] have showcased the utility of deep learning techniques in assisting with lung cancer staging by prognosticating the TNM classification. Furthermore, deep learning methods have been particularly instrumental in identifying tumor dimensions, visceral pleural invasion (VPI), and nodal involvement from preoperative chest CT scans. They have also been harnessed to forecast N2 disease in clinical stage I non-small-cell lung cancer (NSCLC) from chest CT images and predict distant metastasis in ostensibly early-stage NSCLC patients [7]. Additionally, numerous authors have illustrated how the integration of Artificial Intelligence (AI) into medical imaging can enhance the precision of tumor dimension measurements and the detection of nodal involvement, lymph vascular invasion, visceral pleural invasion, and distant metastasis in patients afflicted with non-small-cell lung cancer (NSCLC). 

### 3.1. AI-Based Algorithms for Predicting the “T” Stage

The T parameter classification involves assessing the morphological and dimensional characteristics of the primary tumor, including its location, its extent, and the involvement of neighboring organs or structures, determined through non-invasive diagnostic techniques such as CT and PET scans, as well as evaluating its cytohistological characteristics via invasive diagnostic methods like cell or tumor tissue sampling. Kirienko et al. [8] developed a convolutional neural network (CNN) on a cohort of 472 patients to stage primary lung tumors based on PET/CT images acquired prior to biopsy or surgery. They achieved promising results in classifying T1–T2 and T3–T4 lung cancers with an accuracy of 87%. Their study demonstrated the potential utility of these CNNs in assessing the T parameter in lung cancer, providing rapid determination of whether a primary lung tumor falls into the T1–T2 or T3–T4 categories based on baseline PET/CT images.

Similarly, Weikert et al. [9] devised an AI-based algorithm capable of effectively detecting lung T1 lesions (detection rate: 90.4%) and moderately detecting T2 tumors (detection rate: 70.8%): r 0.908 for T1 (*p* < 0.001) and r 0.797 for T2 (*p* < 0.001). However, the detection rates appear to be influenced by tumor stage, with lower detection rates associated with more advanced stages. Consequently, the algorithm’s performance proved weaker in T3 staging (detection rate: 30%) and poorer in T4 staging (detection rate: 9%), with r 0.520 for T3 (*p* 0.047) and r 0.748 for T4 (*p* 0.053), due to pleural involvement, a significant factor contributing to misidentification. Numerous studies have examined the size of the solid tumor component via analysis of chest computed tomography (CT) images and the size of the invasive component via pathology, using the eighth edition of the American Joint Commission on Cancer (AJCC) staging system. These parameters have been identified as superior prognostic predictors compared to the total tumor size [10]. In most cases, subsolid lung nodules, persistently present in CT chest images, represent pathologically preinvasive lesions, such as atypical adenomatous hyperplasia (AAH) or adenocarcinoma in situ (AIS), or lung cancers such as minimally invasive adenocarcinoma (MIA) or invasive adenocarcinoma [11]. Zhao et al. [12] created a DL model that can differentiate the AAH-AIS group, MIA group, and invasive adenocarcinoma group using data obtained from 651 nodules ≤10 mm in size. Furthermore, they tested their AI-based model, comparing its diagnostic performance with that of four radiologists through an external analysis performed with 128 pathologically proven nodules. Surprisingly, the DL model achieved a higher F1 score (described as 2 × (precision × recall)/(precision + recall)) compared to that of the four radiologists for the task of three-group classification (AAH-AIS vs. MIA vs. invasive adenocarcinoma groups). However, histological subtyping has posed challenges, particularly regarding diagnostic discrepancies between invasive and non-invasive cancer subtypes. AI holds promise in addressing these inconsistencies and facilitating standardization. Choi et al. [13] designed a DL algorithm to predict visceral pleural invasion (VPI), a crucial T2 descriptor with negative prognostic implications. Their algorithm, developed on 676 patients with clinical stage IA lung adenocarcinoma, demonstrated comparable performance to evaluations by three board-certified radiologists, achieving even higher sensitivity and specificity (AUC = 90–100%) (adjusted odds ratio, 1.07; 95% CI, 1.03–1.11; *p* < 0.001). 

### 3.2. AI for Predicting “N” Lymph Node Involvement

The precise delineation of the N factor (lymph node involvement) holds significant importance in determining the course of therapy. The accurate identification of early-stage NSCLC cases devoid of lymphovascular invasion (LVI) or nodal invasion before surgery could enable patients to undergo limited (sublobar) resection. This approach offers substantial advantages, especially for patients with compromised cardiopulmonary function. Long-term results from the Japan Clinical Oncology Group JCOG0201 trial indicate that such patients experience more favorable outcomes compared to those undergoing lobectomy. Limited surgery is recognized to improve prognosis in lung cancer patients with specific characteristics, including ground-glass opacity (GGO) nodules, a consolidation-to-tumor (C/T) ratio of 0.5 or less, and a tumor size greater than 2 cm but not exceeding 3 cm [14]. Accurate N staging holds paramount importance in the optimal treatment and management of non-small-cell lung cancer (NSCLC) patients. PET/CT imaging and endobronchial ultrasound EBUS/TBNB represent the current standard diagnostic tools for N staging. Regrettably, the occurrence of occult N2 metastases is relatively high, affecting up to 8.5% of clinical stage I NSCLC cases. This circumstance has spurred researchers to investigate alternative methods for achieving more precise N staging. Radiomics emerges as a promising approach to enhance the accuracy of N staging in NSCLC patients, leveraging its capacity to analyze extensive datasets of medical images and extract quantitative information that traditional diagnostic methods may overlook. In solid tumors characterized by a consolidation-to-tumor (C/T) ratio of 1.0, the tumor itself emerges as the most critical region for predicting lymphovascular invasion (LVI) or nodal involvement. Conversely, in part-solid tumors with a C/T ratio of less than 1.0, the periphery of the tumor, constituting the interface between the tumor and the adjacent lung parenchyma, emerges as the most significant area for predicting LVI or nodal involvement. This observation suggests the presence of factors that are imperceptible to the human eye but nonetheless influence the invasiveness of a part-solid tumor. To address this, Beck et al. [15], in 2021, integrated a deep learning (DL) tool into preoperative CT images of 600 pathologically confirmed stage I–III NSCLC patients with tumor size ≤ 3 cm who underwent upfront surgery to detect lymphovascular invasion (LVI) or nodal involvement. Unlike lymph node metastasis, the presence of LVI cannot be clinically and preoperatively determined based on CT imaging features. Nonetheless, the incorporation of LVI or nodal metastasis into the prediction of invasiveness using CT images represents a promising advancement. Tumor size and the consolidation-to-tumor (C/T) ratio serve as indicators of nodal invasiveness, both as evaluated via CT and determined by radiologists. 

Their predictive model for LVI/nodal involvement demonstrated a sensitivity of 75.8%, specificity of 67.6%, and accuracy of 70.8%, with an AUC of 0.77 [16]. Recently, Zhong et al. [17] developed a DL model for predicting N2 metastasis and conducting prognostic stratification in clinical stage I NSCLC patients. Utilizing chest CT images from 3096 patients, their model exhibited an AUC of 0.82 in forecasting N2 metastasis in early-stage NSCLC. Furthermore, higher DL scores correlated with poorer overall survival (adjusted hazard ratio, 2.9; 95% CI: 1.2–6.9; *p* = 0.02) and recurrence-free survival (adjusted hazard ratio, 3.2; 95% CI: 1.4–7.4; *p* = 0.007). The authors concluded that deep learning could accurately predict N2 staging and categorize NSCLC patients according to prognostic stage I. On another note, Tau et al. [18] demonstrated that CNN analysis of primary PET images from untreated NSCLC patients accurately predicted the N category with a sensitivity and specificity of 74% and 84%, respectively. The algorithm also effectively predicted nodal involvement (80%) with good diagnostic accuracy but yielded unsatisfactory results in the presence of distant metastases (63%) (accuracy 0.80 ± 0.17, sensitivity 0.74 ± 0.32, specificity 0.84 ± 0.16, AUC 0.80 ± 0.01).

### 3.3. AI for Predicting Distant Metastases “M” at Diagnosis

All patients diagnosed with confirmed or suspected neoplastic disease typically undergo screening, which includes whole-body CT, bone scintigraphy, and, notably, PET-CT, for distant metastases at common sites. Lung cancer commonly metastasizes to several organs, including the liver, brain, bone, and adrenal glands, with radiomics demonstrating remarkable accuracy, especially in identifying adrenal gland metastases. In 2020, Wu et al. [19] devised eight AI models aimed at defining “N” and “M” staging in 1102 NSCLC patients with tumor mass ≤2 cm, utilizing both clinical and radiological features. The majority of these eight models exhibited high AUCs, with all machine learning (ML)-based models achieving AUCs ranging from 0.86 to 0.89. Through the feature selection approach, tumor size, density, SUVmax, and age were identified as the most significant predictive risk factors for nodal involvement and distant metastasis. Coroller et al. [20] investigated the correlation between radiomic data and distant metastasis (DM), as well as overall survival. They conducted a study involving 182 lung cancer patients treated with chemo-radiotherapy, from whom they extracted 635 radiomic features. The aim was to evaluate the prognostic potential of radiomic features as biomarkers for DM. Their findings indicated that the combination of clinical and radiomic features exhibited a significantly higher association with DM (*p*-value = 0.049) compared to the clinical model for patients with locally advanced adenocarcinoma (ADC). This signature could facilitate the early identification of locally advanced patients at risk of developing DM, enabling clinicians to personalize treatment strategies, such as intensifying chemotherapy, to mitigate the risk of DM and enhance overall survival. Huang et al. [21] proposed a machine learning algorithm aimed at identifying the optimal prognostic index for brain metastases within a large patient cohort encompassing various tumor types. The cohort comprised 446 patients in the training set and 254 in the testing set, predominantly with non-small-cell lung cancer (NSCLC) (90.7%). Seven clinical and qualitative features including age, performance status, the presence of extracranial metastases, primary tumor control, the number of lesions, maximum lesion volume, and the administration of chemotherapy were utilized to predict patients’ prognoses. Their study demonstrated that this machine learning-based prognosis outperformed conventional statistical methods in terms of accuracy (88%), sensitivity (92%), and specificity (85%), with an AUC of 0.97.

## 4. Artificial Intelligence and Prognosis

One of the pivotal applications of digital pathology lies in predicting patient prognosis and response to treatment, thereby enabling precision medicine grounded in pathological histomorphological phenotypes. While certain pathological features, such as lung cancer classification and subtypes, have been identified as crucial prognostic factors, establishing a direct correlation between pathological images and survival outcomes remains a significant challenge [13]. 

AI-based analysis of the entire tumor volume prior to treatment has been utilized to predict outcomes in lung cancer, encompassing the control of local and distant cancer spread, as well as the survival of patients undergoing various treatment modalities for non-small-cell lung cancer (NSCLC), such as surgical procedures, radiotherapy, chemotherapy, targeted molecular therapy, or immunotherapy. AI holds promise as a non-invasive approach for follow-up examinations by analyzing radiological follow-up CT/PET images and H&E-stained slides and integrating data from innovative sources like liquid biopsies. These liquid biopsies provide insights into the genetic phenotype of lung cancers without requiring a tissue re-biopsy for confirming disease progression. A study examined 2186 full-scan images of paraffin tissue sections from lung adenocarcinoma and lung squamous cell carcinoma sourced from TCGA, alongside 294 images from the Tissue Microarray (TMA) database [18]. 

Quantitative features were extracted from 9879 images, and the machine learning algorithm identified the top features. The results indicated that these features could effectively predict the survival times of patients with lung adenocarcinoma (*p* < 0.003) and lung squamous cell carcinoma (*p* < 0.023). Furthermore, data from the TMA database were utilized to validate the accuracy of the assessment model. The analysis revealed a statistically significant difference in the prediction accuracy between these two tumor types (*p* < 0.036), suggesting that the automatically acquired pathological image features accurately predicted the prognoses of lung cancer patients [18]. Radiomics theory postulates that radiographic phenotypes are indicative of fundamental pathophysiological changes that allow for the prediction of tumor response and prognosis [22]. Prior studies have indicated that volumetric measures derived from RECIST and World Health Organization (WHO) criteria can be employed to evaluate the agreement of automatically segmented lung lesions. Nevertheless, these measures seem to have constraints in effectively characterizing the complex nature of tumors [23]. To address this limitation, Balagurunathan et al. [24] proposed various categories for characterizing primary lung cancers with diverse characteristics. These categories included size metrics such as volume, diameter, and border length, as well as shape descriptors like shape index, compactness, and asymmetry. Additionally, features related to the border region (e.g., border length and spiculation), related to the lung field, image intensity-based attributes (e.g., mean, standard deviation, average airspace, airspace deviation, energy, entropy, skewness, among others), and transformed texture descriptors (utilizing wavelet transform, including entropy, energy, and Laws features) were considered. These repeatable features were employed to forecast the prognostic score of a conventional radiologist, yielding an AUC of 0.9. This study concluded that these findings would facilitate the identification of reproducible, informative, independent, and prognostic imaging biomarkers capable of predicting response to therapy.

### 4.1. AI for Predicting Prognosis and Tumor Recurrence after Surgery

Previous radiomics investigations have often omitted patients who have undergone surgical intervention for lung cancer. This exclusion is grounded in the presumption that predicting tumor response solely based on its phenotype becomes irrational once the tumor has been surgically removed [20]. However, a study by Hosny et al. [25], in 2018, involving 1194 patients with non-small-cell lung cancer (NSCLC) proposed that convolutional neural networks (CNNs) could be beneficial in predicting 2-year overall survival in surgically treated patients using CT data. This study also illustrated each CNN’s capability to categorize patients into high- and low-mortality-risk groups. CT imaging is preferred due to its relatively stable radiodensity compared to other modalities such as MRI and PET. Furthermore, this study suggests that deep learning (DL) features may offer superior prognostic value for surgically treated patients. Surgical resection remains the gold standard treatment for early-stage lung cancer. Artificial Intelligence (AI) has been utilized in preoperative evaluation and prognostication following surgery, thereby aiding the identification of patients who may benefit from adjuvant chemotherapy post-surgery (see Table 1) [26]. In preoperative assessment, radiologist-level AI could help to predict visceral pleural invasion [13] and identify early-stage lung adenocarcinoma suitable for sublobar resection [27]. Following surgery, AI could contribute to predicting prognosis. Models utilizing radiomic feature nomograms could discern high-risk groups characterized by a postoperative tumor recurrence risk 16 times higher than that of the low-risk group [28]. The convolutional neural network (CNN) model, pre-trained with the radiotherapy dataset, effectively forecasted 2-year overall survival post-surgery [25]. The model integrating genomic and clinicopathological features was able to identify patients at risk of recurrence and suitable for adjuvant therapy [26]. Pathology images captured the high-resolution histomorphological details of tumors. Identifying and describing tumor regions in pathology images manually is time-consuming and subjective. In 2017, Wang et al. [29] developed a machine learning (ML) model to forecast the risk of recurrence in early-stage non-small-cell lung cancer (NSCLC) using digital H&E tumor microarray (TMA)-stained slides from surgically excised tissue samples. Their findings revealed that a combination of nuclear shape, texture, and architectural features served as predictive indicators of recurrence in early-stage NSCLC, irrespective of clinical parameters such as gender, cancer stage, and histological subtype. The model demonstrated an accuracy of 81% in predicting recurrence within the training group and 75% within the validation group. Notably, the model’s predictions were identified as an independent prognostic factor. Consequently, it could potentially serve as a decision support tool to assist in determining the utilisation of additional treatment in early-stage lung cancer. Although the model was only assessed in a limited cohort of 235 patients, the concept holds promising implications for the future [30]. Similarly, a 2018 study trained a convolutional neural network (CNN) model to automatically extract histopathological features of lung adenocarcinoma (ADC) [30]. The AI system successfully detected tumor-related features in pathology images and developed a model for predicting recurrence that remained independent of other clinical variables. With an overall prediction accuracy of 89.8%, the CNN model demonstrated strong performance. Furthermore, the patient prognostic model, trained on the NLST cohort, underwent independent validation in the TCGA cohort, showcasing the model’s generalizability and applicability to diverse lung adenocarcinoma patient cohorts. Song et al. [31] demonstrated an association between features derived from CT images and overall survival in patients with non-small-cell lung cancer (NSCLC). Their findings suggested that tumor heterogeneity quantified through CT phenotypic signatures might indirectly indicate tumor prognosis. Detecting poor prognosis via non-invasive methods could potentially mitigate unnecessary drug toxicity and costs, thereby enabling the more precise selection of treatment regimens. These findings imply that AI has the potential to become robust and widely applicable for practical use in clinical care. Furthermore, it could serve as a non-invasive and cost-effective routine medical test, thereby contributing to improved outcomes. These studies underscore AI’s significant potential as a prognostic tool. AI has the capacity to offer a non-invasive alternative for pathological diagnosis and enhance the accuracy of therapeutic treatment decisions by alleviating the workloads of radiologists, pathologists, and oncologists.

### 4.2. AI for Predicting Response and Prognosis after Chemotherapy, Targeted Therapy, and Immunotherapy

In a recent investigation by Khorramin et al. [32], CT-based radiomic features were derived from the peri- and intertumoral tissue of lung adenocarcinoma in 125 patients undergoing pemetrexed-based platinum chemotherapy at the Cleveland Clinic. They demonstrated the capability of AI to forecast the response to chemotherapy, achieving a receiver operating characteristic (ROC) area under the curve of 0.82. The study concluded that radiomics exhibited greater overall effectiveness in anticipating high-risk patients for treatment compared to traditional clinicopathological assessments, and this association correlated with both time to progression and overall survival in NSCLC patients. Aerts et al. [33] enrolled 1019 patients to extract 440 radiomic features for predicting EGFR mutations linked to gefitinib responsiveness, aiming to elucidate the inter-relation between radiomics and mutation status at baseline, particularly concerning changes in therapy among patients with and without EGFR mutations. They found that radiomics could non-invasively define a gefitinib response phenotype (AUC = 0.67, *p* = 0.03) capable of distinguishing between tumors with and without EGFR mutations at baseline. Moreover, it could predict sensitive and resistant responders during follow-up with minimal additional expense, as imaging is routinely and repeatedly utilized in clinical practice. Mu et al. [34] investigated 194 patients with histologically confirmed stage IIIB-IV NSCLC using pre-treatment PET/CT imaging, developing radiomic models to forecast which NSCLC patients would derive the most benefit from immunotherapy. Additionally, this radiomics signature was combined with ECOG status and histology for the personalized prediction of progression-free survival (PFS) and overall survival (OS) prior to the initiation of checkpoint blockade immunotherapy. Their findings suggested that an effective and consistent radiomic approach could serve as a predictive biomarker (AUC = 0.82) for immunotherapy response. Furthermore, radiomics could estimate PFS and OS, offering potential real-time support for more precise and individualized decision-making in the immunotherapy-based treatment of advanced NSCLC patients (see Table 2).

### 4.3. AI for Predicting Response and Prognosis after Radiotherapy for NSCLC

Radiomics signatures have promising potential in predicting therapy effectiveness for patients [35]. In instances where patients with non-small-cell lung cancer (NSCLC) are medically unfit for surgery or decline it, stereotactic ablative radiotherapy (SABR) or stereotactic body radiotherapy (SBRT) is recommended (see Table 3). The elevated doses administered in SBRT have yielded local control rates of up to 90% at three years post-treatment, comparable to those observed after surgery [36]. However, radiation-induced lung injury (RILI), such as radiation fibrosis, may manifest following SBRT. Some benign changes may resemble tumor recurrence in size and shape on routine CT scans conducted every three months during follow-up. Mattonen et al. [37] developed a radiomics algorithm with high predictive accuracy for distinguishing between recurrence and fibrosis (AUC = 0.7). This advancement enables the timely administration of salvage therapies to patients and reduces the risk of unnecessary invasive biopsy procedures for those with benign fibrosis. Additionally, this automated decision support system enhances physician assessment of response to SBRT, facilitating the early detection of recurrence, twenty times faster than manual segmentation. Fave et al. [36] conducted a study to investigate whether radiomic features of NSCLC change during therapy and can enhance prognostic models. They analyzed features from pre-treatment and weekly intra-treatment CT images of 107 patients with stage III NSCLC. The study revealed significant changes in all radiomics features during radiotherapy, indicating that radiomics cannot predict loco-regional recurrence or distant metastasis. Nevertheless, these features remained prognostic for overall survival. Other studies have developed predictive models based on CT [34] and PET/CT [38] to assess the risk of distant metastases in NSCLC patients undergoing SBRT. The 2018 study by Oikonomou et al. [38], involving 150 patients with 172 lung cancers treated with SBRT, demonstrated that radiomics remained the sole predictor of overall survival (OS), disease-specific survival (DSS), and regional control. The study concluded that radiomics based on PET/CT provided complementary information for predicting control and survival in SBRT-treated lung cancer patients. A larger study in 2019 by Lou et al. [39], including 849 patients treated with SBRT for stage IA to IV NSCLC, concluded that their deep learning (DL) model exhibited predictive characteristics that could assist in personalizing the radiation dose by combining clinical variables with radiomics using CT images. Nemoto and colleagues [40] constructed two neural networks (NNs) for the prediction of overall survival (OS) and cancer progression in the first 5 years after SBRT, which were evaluated using both internal and external test datasets. The survival and cumulative incidence curves showed significant stratification. NNs identified low-risk cancer progression groups of 5.6%, 6.9%, and 7.0% in the training, internal test, and external test datasets, respectively, suggesting that 48% of patients with peripheral Tis-4N0M0 NSCLC could be categorized as being of low risk for cancer progression.

### 4.4. AI for Predicting Treatment Strategy/Treatment Decision

Accurate prognosis prediction in lung cancer patients is crucial for clinicians to assess tumor progression and devise suitable treatment strategies. Traditionally, prognosis and subsequent treatment decisions have relied heavily on cancer staging. However, patients with the same cancer stage often exhibit diverse prognoses due to varying responses to treatment [42]. Moreover, the rapidly evolving landscape of medical oncology may render it challenging for physicians to stay abreast of the latest therapeutic guidelines. To address these challenges, researchers have explored the application of deep learning (DL) techniques to predict personalized treatment strategies for individual patients. One notable example is the Watson for Oncology (WFO) system, developed in the USA and trained using data from the Memorial Sloan Kettering Cancer Center (MSKCC) incorporating the latest evidence and guidelines. WFO assists clinicians in offering accurate, tailored treatment plans for cancer patients by extracting valuable insights from medical records. By inputting tumor-related data manually, WFO can predict survival likelihood and offer personalized treatment recommendations for individual patients [43].

## 5. Limitations of AI-Based Models

The future of applying AI to lung cancer may focus on integration and practical applications. One approach is to integrate small datasets to create larger training datasets, as AI is driven by data. However, data sharing regulations are a major challenge for researchers. One potential solution is federated learning, where trained parameters are shared instead of raw data [44,45]. In this method, models are trained separately at different hospitals, and only the trained models are sent to a central server, avoiding direct access to raw data. The final model is then returned to the individual hospitals. Another aspect to consider is the integration of different disciplines and data sources into lung cancer research, including radiology, pathology, demographic and clinical data, as well as old and new technologies. Such integration can provide a more complete picture of reality and aid in the construction of predictive models [46,47] (Figure 1). This concept gives rise to the idea of multi-omics or ‘medomics’ [48,49], reflecting the multidisciplinary teams in lung cancer clinical care [50]. The pursuit of a combination of different domain knowledge and multidisciplinary integration holds promise for the future. In addition to improving model accuracy through larger training sample sizes and multidisciplinary integration, the practical application of AI software poses challenges. Although studies have shown promising results in applying AI to lung cancer and some products have received FDA approval [51,52,53,54,55,56], real-world implementation in clinical workflows remains uncommon. Barriers such as user interface, the speed of data analysis, the scalability of AI programs, internet bandwidth, and resource requirements hinder widespread adoption. More infrastructure needs to be developed before we can fully embrace an AI-enabled world.

## 6. Limitations of AI-Based Models and Future Directions

The current models mostly depend on retrospective study, which inevitably caused biases such as selection bias and information bias. Therefore, prospective studies are urgently needed to verify the efficiency of these models. More generally, since reproducibility is one the main criticisms of this new era, all AI models that will be developed in the future should be tested across diverse geographic regions, populations, and healthcare settings to determine their general effectiveness and robustness. Increasing access to AI requires lifting or reducing some financial or regulatory obstacles, so as to increase availability in less privileged healthcare systems. Roadblocks regarding ethical and legal considerations must be also addressed. Moreover, AI models must be simplified to increase utility among relatively technologically illiterate populations. 

Although AI has shown promising results in medical image analysis, it has some limitations due to the need for labeled data provided by radiologists, which may reflect the limitations of human perception and analytical discrimination. One of the main challenges in applying deep CNNs to medical images is the resolution of the images [56]. While AI models have demonstrated comparable or superior performance to humans, the complexity of these models makes it difficult to interpret and understand how they make to their decisions, which has led to the conception of AI models as “black boxes” [57]. Another major concern is the generalizability of these models to all patients, which could be addressed by developing continuous learning systems that utilize cloud techniques to allow for the real-time delivery of clinical records and continuous modification of the underlying training models. This would ensure the machine-independent reproducibility of the models [58].

The evolving role of AI-based models for precision medicine in NSCLC will implement radiomics and liquid biopsies (circulating tumor cells and/or nucleic acids detection) to achieve robust data on tumor biology, disease progression, and response to treatment in a longitudinal non-invasive fashion [58]. 

Future applications of AI for precision medicine in NSCLC may implement the integration of multi-omics data to produce AI-based algorithms for treatment decisions and prognostic predictions (see Figure 2). The integration of larger and more diverse datasets may overcome the current limitations of AI use. In this framework, our Research Team is working on developing lung cancer multi-omics digital human avatars for integrating precision medicine into clinical practice (LANTERN-project [57]). This is an interesting example of the application of integrated AI techniques; by meticulously collecting genomic, radiomic, and metabolomic profiles, alongside comprehensive clinical and treatment data, the study aims to construct Humanized Digital Avatars (HDAs) that dynamically represent each patient’s unique molecular and clinical landscape. This approach not only promises to optimize patient-specific treatment strategies but also holds the potential to redefine the future of cancer care by contributing to the ongoing transformation fueled by advanced machine learning and AI techniques.

Precision medicine enables the development of individualized treatment approaches tailored to the specific needs and risk factors of each patient. This can help to improve screening strategies, reduce adverse events, and ultimately enhance quality of life for patients. From this viewpoint, the importance of the collaboration of different healthcare professionals, to integrate the results from AI into a real-life setting, becomes clear.

Technological advancements such as big data, Artificial Intelligence, machine learning, and predictive analytics play a crucial role in cancer treatment. For this reason, in the near future, collaboration between engineers, biologists, statistics, informatics and technology companies, and healthcare organizations will be a day-to-day practice to enhance AI-based results for every patient.

These concepts enable more precise and personalized predictions and support physicians in the treatment and monitoring of their patients.

## 7. Conclusions

Artificial Intelligence (AI) has largely changed the overall management of non-small-cell lung cancer (NSCLC) by enhancing different aspects, including staging, prognosis assessment, treatment prediction, response evaluation, recurrence/prognosis prediction, and personalized prognostic assessment. Indeed, AI-based technologies, despite their infancy, have gained great attention within the oncology community as they could potentially foster the optimal, personalized management of cancer patients. Indeed, by tackling the complexity of the highly heterogeneous NSCLC disease, AI approaches will pave the way for a paradigm shift in the field of informed, data-driven clinical decisions in the near future.

Collaboration among clinicians, data scientists, and regulators is critical for the responsible implementation of AI and for maximizing its benefits in providing more personalized prognostic assessment.

Several challenges still remain, though their prospective validation within a large number of institutions over diverse populations will ultimately lay the foundation for their real-world implementation.

## Figures and Tables

**Figure 1 cancers-16-01832-f001:**
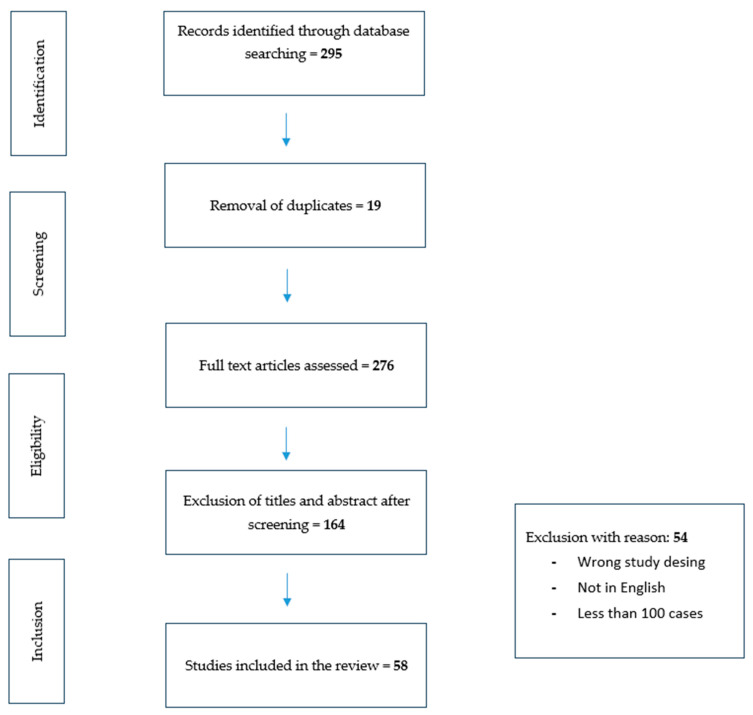
Preferred Reporting Items for Systematic Reviews and Meta-Analysis flow diagram for the literature search.

**Figure 2 cancers-16-01832-f002:**
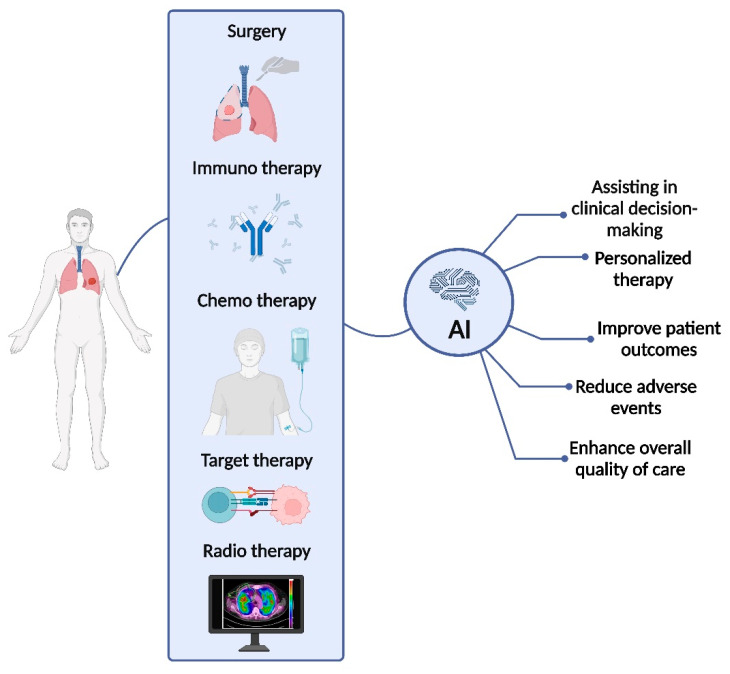
AI applications for personalizing therapy and predicting response and prognosis in NSCLC patients (Created with BioRender.com, www.biorender.com accessed on 20 January 2024).

**Table 1 cancers-16-01832-t001:** The role of AI predicting NSCLC prognosis and tumor recurrence after surgery/SBRT.

Study	Number of Patients	Prediction	Results
Hosny et al. [25] 2018	1194	This study explores deep learning applications in medical imaging, allowing for the automated quantification of radiographic characteristics and potentially improving patient stratification.	The CNN was able to significantly stratify patients into low- and high-mortality-risk groups in both the radiotherapy (*p* < 0.001) and surgery (*p* = 0.03) datasets.
Wang et al. [29] 2017	Retrospective cohort of early-stage NSCLC (I–II) patients (Cohort #1, *n* = 70; Cohort #2, *n* = 119 and Cohort #3, *n* = 116).	They trained an ML model to predict the risk of recurrence in early-stage NSCLC based on digital H&E tumor microarray (TMA)-stained slides of surgically excised tissue samples.	The combination of nuclear shape, texture, and architectural features was predictive of recurrence in early-stage NSCLC, independent of clinical parameters such as gender, cancer stage, and histological subtype.
Wang et al. [30] 2018	389	They trained a deep CNN model to automatically extract histopathological features of lung ADC.	The patient prognostic model was trained using the NLST cohort and independently validated in the TCGA cohort, demonstrating the applicability of the model to other lung ADC patient cohorts.
Song et al. [31] 2016	661	The identification of poor prognosis via non-invasive methods.	They demonstrated an association between features extracted from CT images and overall survival in NSCLC patients. They concluded that tumor heterogeneity quantified via CT phenotypic signatures may indirectly reflect tumor prognosis.

**Table 2 cancers-16-01832-t002:** The role of AI in response to systemic therapy for NSCLC.

Study	Number of Patients	Prediction to Response to	Results	Comments
Khorrami et al. [32] 2019	125	Chemotherapy: pemetrexed-based platinum chemotherapy	The radiomics signature was significantly associated with the following: Response to chemotherapy: AUC of 0.82 ± 0.09 Time to progression: HR 2.8; 95% CI: 1.95, 4.00; *p* < 0.0001) Overall survival HR 2.35; 95% CI: 1.41, 3.94; *p* = 0.0011)	The results from the training set were confirmed in the independent validation set.
Aerts et al. [33] 2016	47	Gefitinib in early-stage adenocarcinoma	Radiomics-feature Laws-Energy was significantly predictive for EGFR mutation status (AUC = 0.67, *p* = 0.03)	Capacity to predict EGFR mutations for non-invasive diagnosis
Mu et al. [34] 2020	194	Anti-PD-(L)1 immunotherapy	Multiparametric radiomics signature was able to predict the following: Durable clinical benefit with AUCs of 0.86 in the retrospective test and 0.81 in the prospective test cohorts. Progression-free survival in the training (*p* < 0.001), retrospective test (*p* = 0.001), and prospective test cohorts (*p* < 0.001), Overall survival in the training (*p* < 0.001), retrospective test (*p* = 0.002), and prospective test cohorts (*p* = 0.002)	IIIB-IV NSCLC with pre-treatment PET/CT images

AUC: area under the curve; HR: hazard ratio; CI: confidence interval; NSCLC: non-small-cell lung cancer.

**Table 3 cancers-16-01832-t003:** The role of AI in response to radiotherapy.

Study	Number of Patients	Prediction to Response to	Results	Comments
Huynh et al. [35] 2016	113	SABR Prescribed radiation dose (Gy): 54 (18–60) Radiation dose per fraction (Gy): 18 (10–18)	Radiomics features were able to predict overall survival, cancer-specific survival, and distant metastases development.	Stage I–II NSCLC 15 imaging features (3 conventional and 12 radiomic features) and 4 clinical parameters (age, gender, performance status, overall stage) were included in the analysis.
Mattonen et al. [37] 2015	22	SABR	Radiomics able to distinguish post-SABR fibrosis from tumor recurrence (AUC 0.70).	Study validated considering manual and automatic segmentation.
Fave et al. [41] 2017	107	Radiotherapy (66 or 74 Gy) and concurrent chemotherapy	Three prognostic models were studied: 1: Only clinical variables; 2: Clinical variables and pre-treatment radiomics features; 3: Clinical variables, pre-treatment radiomics features, and changes in radiomics features between pre- and post-treatment imaging. Creating prognostic models with pre-treatment radiomics features (2) and changes in radiomics features between pre- and post-treatment imaging (3) permit better stratification for overall survival, disease-free survival, and distant metastases development.	Stage III NSCLC.
Oikonomou et al. [38] 2018	150	SABR Total dose: 48–56 Gy	Prognostics models predictive for the following were studied: - Model 1: Recurrence-free survival (without considering SUV) (*p* = 0.04) and distant control (*p* = 0.01); - Model 2: Disease-specific survival (with SUVmax) (*p* = 0.03); - Model 3: Overall survival (without and with SUVmax, *p* = 0.02 and *p* = 0.02) and disease-specific survival (without and with SUVmax, *p* = 0.02 and *p* = 0.02). - Model 4: Overall survival (without and with SUVmax, *p* = 0.004 and *p* = 0.004), distal control (without and with SUVmax, *p* = 0.02 and *p* = 0.02), and regional control survival (without and with SUVmax, *p* = 0.02 and *p* = 0.02).	Radiomics applied on PET/CT. Four predictional models including different radiomics features, including or excluding the SUV value.
Lou et al. [39] 2019	849	SABR 50–60 Gy in 3–5 fractions	Deep profiler score generated from deep profiler signatures. Predictive for 3-year local failure: 5.7% in low-risk group vs. 20% in high-risk group (*p* < 0.001).	Multivariable models including deep profiler and clinical variables predicted treatment failures with a C-index of 0.72 (95% CI: 0.67–0.77), which was a significant improvement when compared to classical radiomics (*p* = <0.001) or 3D volume (*p* = <0.001).
Nemoto et al. [40] 2022	692 (study group) + 100 external validation set	Predictions of SBRT outcomes using artificial neural networks	The survival and cumulative incidence curves were significantly stratified in Tis-4N0M0 NSCLC patients who underwent SBRT for curative intent.	

AUC: area under the curve; HR: hazard ratio; CI: confidence interval; NSCLC: non-small-cell lung cancer.

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
