# Peer review of "Implementation of Artificial Intelligence in Personalized Prognostic Assessment of Lung Cancer: A Narrative Review"

_cancers, 2024, doi:10.3390/cancers16101832_

Round 1

Reviewer 1 Report

Comments and Suggestions for Authors

I have the following comments:

1) The Introduction provides broad and quite exhaustive information on the basics of AI and its applications in healthcare, but it should also introduce in more detail which are the unmet needs of clinicians in the diagnostic and therapeutic management of NSCLC that AI could meet. Please reshape the Introduction on this basis.

2) While the authors stressed that this is a narrative review, I suggest (if possible) that it be transformed into a systematic review by reporting in detail the exact search criteria entered for Pubmed searches, how many papers were retrieved, the proportion of those discarded (and why), and the total number of papers that were finally evaluated. These steps should be summarized in a dedicated PRISMA flow chart. 

In any case, please report the degree of experience (e.g., in terms of years after board certification or of subspecialty professional activity) and qualifications of the authors who performed literature search and analysis.

3) Lines 88-89. What did you mean by ‘the beginning’? Please enter a specific date.

4) Line 122. Please remove the underline.

Comments on the Quality of English Language

Some minor English editing should be performed.

Author Response

A detailed file resuming point by point revisions is attached

Reviewer 2 Report

Comments and Suggestions for Authors

The overall review is very well articulated with relevant updates , however i have the following suggestions which can be implemented to improve the review .

Here are my suggestions : 

  1. Clarify Methodology and Expand on Statistical Analysis:

    • The methodology section could be improved by providing more details on the data collection processes and the specific types of AI and machine learning models used. Include a discussion on the validation techniques and any cross-validation methods if employed. Clarifying these aspects would strengthen the reliability of the research findings.
    • Include more comprehensive statistical analysis results to support the findings. Details such as confidence intervals, p-values, and the statistical significance of the results would help in assessing the robustness of the AI models discussed.
  2. Enhance Discussion on Limitations and Future Research:

    • Expand the discussion on the limitations of the current AI models used in lung cancer prognostic assessment. Address potential biases and the generalizability of AI predictions to different populations or clinical settings.
    • Propose future research directions more clearly. Suggestions could include the integration of larger and more diverse datasets, the exploration of new AI techniques, or the implementation of real-time AI systems in clinical practice. Discuss how these advancements could address current limitations.
  3. Increase Interdisciplinary Collaboration:

    • Emphasize the need for increased collaboration between oncologists, computer scientists, and data analysts. This interdisciplinary approach is crucial for developing more accurate and clinically applicable AI tools.
    • Discuss the potential for partnerships with technology companies and healthcare organizations to facilitate the development and testing of AI models in real-world settings. This would not only enhance the article's practical implications but also underline the importance of collaborative efforts in advancing AI applications in healthcare.

These enhancements would not only improve the depth of the review but also increase its practical relevance and applicability in the field of lung cancer treatment and prognosis.

Author Response

(The authors gave the same response as above.)

Round 2

Reviewer 1 Report

Comments and Suggestions for Authors

Thank you for your reply.

At line 120, the term 'PRIMA' should be replaced with 'PRISMA'.

Comments on the Quality of English Language

The English language has improved, but some minor issues remain.